# COVID-19 and University Students’ Well-Being: An Ecological and Multidimensional Perspective on Post-Pandemic Effects

**DOI:** 10.3390/bs14100938

**Published:** 2024-10-14

**Authors:** Ciro Esposito, Barbara Agueli, Caterina Arcidiacono, Immacolata Di Napoli

**Affiliations:** 1Department of Humanities, University of Foggia, 71122 Foggia, Italy; 2Surgical, Medical and Dental Department of Morphological Sciences Related to Transplant, Oncology and Regenerative Medicine, University of Modena and Reggio Emilia, 41121 Modena, Italy; aguelibarbara@gmail.com; 3Department of Humanities, University of Naples Federico II, 80138 Napoli, Italy; caterina.arcidiacono@gmail.com (C.A.); immacolata.dinapoli@mail.com (I.D.N.)

**Keywords:** multidimensional well-being, university students, COVID-19 pandemic, ecological perspectives

## Abstract

In February 2020, the Italian government started to adopt measures to contain the spread of COVID-19. This emergency had a strong impact on people’s lives and daily activities, negatively affecting their well-being. One of the groups of people that suffered the most from the pandemic emergency and the related isolation was university students. Based on these considerations, this article analyzes the effects of COVID-19 on Italian students’ well-being during three periods: the first lockdown (March–April 2020), one year later (March–April 2021), and two years after the lockdowns (March–April 2022). Three samples comprising a total of 765 participants (M = 21 years, SD = 2.87) completed an online self-report questionnaire, which included the I COPPE scale (its short form), a tool that measures the perception of present and future well-being, both as an overall evaluation and its six specific domains: interpersonal, community, occupational, physical, psychological, and economic. The results indicated a general trend in the well-being levels of university students from the beginning of the pandemic to 2022. Compared to 2020, in 2021, there was a sharp decline in well-being, whereas in 2022, there was an increase in well-being levels. Practical implications, limitations, and future recommendations arising from the present study are extensively discussed.

## 1. Introduction

In February 2020, the Italian government started to adopt several measures to contain the spread of COVID-19. This widely known emergency had a strong impact on people’s lives and daily activities, such as work, school, free time, and interpersonal relationships. In this context, scholars have attempted to understand the effects of the pandemic and the related containment measures on the perception of well-being and subjective well-being [1]. In particular, studies have considered the different components of well-being. Some studies investigated the emotional dimension, such as negative and positive feelings [2,3,4], states of instability, boredom and loneliness, depression, anxiety, and distress [5]. Others considered the general level of satisfaction [6].

The literature indicates that, from a psychological point of view, people experienced different stressful conditions during the lockdown period due to health concerns, job insecurity, and work–family conflicts [7,8,9].

Furthermore, studies conducted during the first lockdown (March–April 2020) found an increase in feelings of uncertainty and loneliness among people [10,11,12]. In particular, this perception of uncertainty was related to the disruption of daily routine among young people [3,13], as well as to an increase in the sense of helplessness connected to the difficulty of facing the perceived danger [14].

However, it was also noted that the lockdown did not lead to an indiscriminate decline in the levels of well-being, and personal and social factors that helped protect well-being were also analyzed.

Specifically, studies on the well-being of young people found that mental health was mainly associated with the ability to continue with daily activities, the quality of relationships, and how the significant adults in their lives coped with the isolation [15]. In particular, there was an increase in online communication among young people as a response to the malaise caused by the health emergency [16].

Building on the initial evidence provided by the recent literature on the effects of the COVID-19 pandemic, the present study aimed to detect the trend of well-being in its various dimensions among Italian students in the years of the COVID-19 pandemic, i.e., from 2020 to 2022.

Specifically, this study used an ecological and multi-dimensional perspective. As theorized by Isaac Prilleltensky [17,18], the main assumption of this perspective is that well-being is shaped by the interconnection of different domains in such a way that an alteration in one of these domains induces a change in the level of satisfaction in the others.

In complex situations, like the COVID-19 outbreak, people often experience significant emotional consequences and changes in their perception of well-being. To understand these human experiences, the ecological perspective provides a valuable framework for evaluating the impact of contextual factors.

This approach provides a complex, multi-level view of the influences that shape an individual’s well-being and considers the dynamic interaction between an individual and the different components of their environment (physical, social, cultural, and economic). During the pandemic, well-being was affected by a wide range of factors that can be analyzed precisely through this ecological lens. Indeed, during the COVID-19 pandemic, environmental circumstances affected people’s well-being, and people’s reactions also influenced their environment. For instance, the anxiety and stress accumulated during the pandemic changed how individuals interacted with their community and loved ones, impacting social dynamics on a broader scale.

The model proposed by Prilleltensky [19] called I COPPE has already been validated and widely used in Italy [20,21]. It offers a multidimensional vision of well-being that considers its different aspects. It considers both overall individual well-being—understood as a positive state of affairs perceived by the person—and six specific domains of well-being: interpersonal, community, occupational, physical, psychological, and economic.

### University Students and COVID-19 Pandemic

Reflecting on university students’ specific roles and unique contexts is necessary to understand why the COVID-19 pandemic is particularly relevant to this population.

University students face multiple challenges related to their well-being because they are in a critical transitional phase of life, where responsibilities increase and they work on building their identities and careers. During this time, students encounter various challenges, including making independent decisions about their personal and academic lives, adapting to the demands of a less structured learning environment, completing exams and coursework [22], dealing with academic workload [23], uncertainty about the future [24], and financial pressure, and managing social relationships [25]. For many of them, this transition also involves leaving home and moving away from their families for the first time [26].

All these factors create a scenario where their physical and mental well-being can become vulnerable to influence.

The advent of COVID-19 further exacerbated these challenges. The pandemic caused sudden changes, such as the closure of universities and the shift to distance learning, which increased social isolation. Students suddenly had to cope with distance learning, which involves stressful and demanding requirements to meet academic expectations [27]. In particular, distance learning presented relational difficulties in peer interactions and with self-perception [28].

Previous studies have suggested that younger people worried more about the pandemic, and students, mainly, were more distressed than other groups [29], with higher levels of anxiety, depression, and perceived stress. University students experienced strong negative emotions, including worry, stress, fear, anxiety, and depression [30,31,32].

Moreover, recent studies comparing well-being across different countries have shown that Italian students exhibited the lowest levels of well-being and the highest rates of mental health issues [33]. This could be explained by considering the duration and severity of the pandemic-related measures experienced in Italy, which negatively impacted the country [34] and, as previously mentioned, the social and academic pressures that Italian university students face further exacerbated this situation in the university environment [35].

Beyond the most well-known influencing factors and the profound short-term impact that the COVID-19 pandemic has had on the well-being of students worldwide, it is crucial to understand whether and what the long-term effects might be due to prolonged exposure to the pandemic.

The long-term consequences of COVID-19-related measures on the mental health of young people may have left lasting effects, still detectable today [33]. For this reason, it is essential to increase awareness of the pandemic’s long-term impact, even though the most acute and severe phase of COVID-19 has ended.

## 2. Method

### 2.1. Participants and Procedure

This research was conducted over three consecutive years. The data were collected with groups of university students at the University of Naples Federico II, at three different times: during the lockdown (March–April 2020), one year after the lockdown (March–April 2021), and two years after the first lockdown (March–April 2022).

In all three data collection phases, students were recruited through snowball sampling. In particular, an initial group was recruited during the Community Psychology course. Then, the first participants were asked to recruit additional participants, starting from their relational network.

This procedure involved a total of 765 participants with a mean age of about 21 years (SD = 2.87), distributed as follows in the three rounds: 293, 344, and 128 participants, collected in 2020, 2021, and 2022, respectively. Although they included different participants, the three samples were homogeneous for all demographic variables examined, such as age, biological sex, sexual orientation, and housing status (see Table 1).

### 2.2. Measures

Participants completed an online self-report questionnaire that included a socio-demographic section and the I COPPE scale short form [21]. This scale assesses overall well-being through six specific dimensions: interpersonal, community, occupational, physical, psychological, and economic well-being. It consists of 14 items, with 2 dedicated to each dimension. The first item assesses present well-being, whereas the second measures future well-being. Participants indicated their perceived level of well-being for each item using a Cantril scale, which ranges from 0 (the lowest) to 10 (the highest).

### 2.3. Data Analysis

The data were analyzed using SPSS statistical software version 28.

First, a confirmatory factor analysis (CFA) was carried out to verify the structure of the scale. Then, the reliability of the model was evaluated through the Composite Reliability (CR) index. CR values above 0.70 are considered a sign of good reliability [36].

To compare the levels of well-being reported by students over the three years, a univariate analysis of variance (ANOVA) was performed, which allowed for the investigation of statistically significant differences between the mean scores of 2020, 2021, and 2022 for each well-being dimension.

Finally, given the omnibus test nature of the ANOVA, post hoc tests were carried out to pairwise compare the means of the well-being scores in the different years. In particular, the Tukey test and the Dunnet test were used. In the last case, the control group selected was the 2020 sample.

## 3. Results

The CFAs showed a good fit of the 7-factor structure of the I COPPE scale in all three samples.

As shown in Table 2, all factor loadings were significant at an alpha level of 0.01, demonstrating strong inter-item reliability values (R^2^).

Moreover, the Composite Reliability (CR) indices demonstrated a good level of reliability (>0.7) for all dimensions of the I COPPE scale in all three samples (see Table 2).

Regarding the ANOVA, as reported in Table 3, Fisher’s F values were significant for all dimensions of well-being. In particular, the differences between the samples in levels of economic well-being were statistically significant with an alpha level of 0.05, whereas the level of significance of the differences between the means in all other dimensions was 0.001.

These results indicate the existence of a clear difference in the levels of well-being reported by students in the three years investigated.

To further investigate these differences, post hoc tests were conducted (Tukey’s test and Dunnet test) which compared, pair to pair, the means of the well-being dimensions of 2020 with those of 2021, the means of 2021 with those of 2022, and finally the means of 2020 with those of 2022.

The results of the Tukey test and the Dunnet test both indicate a wave trend in well-being levels in almost all dimensions, from 2020 to 2022 (Table 4).

Specifically, the difference between the means (Δ_M_) of the levels of interpersonal well-being shows a significant decline (*p* < 0.001) from 2020 to 2021 (Δ_M_ = −0.57), whereas in 2022, it is higher than that in 2021, but not as high as that in 2020. However, these last two differences are not statistically significant.

A similar trend occurs for almost all dimensions of well-being. The differences between the 2021 and 2020 means are also negative and significant (*p* < 0.001) for community (Δ_M_ = −0.89), occupational (Δ_M_ = −0.85), physical (Δ_M_ = −0.76), psychological (Δ_M_ = −0.85), and overall (Δ_M_ = −0.78) well-being. Even for all these dimensions, there are no significant differences when comparing the means of 2022 with those of 2020.

However, significant positive differences were found when comparing the means of 2022 with those of 2021. Specifically, we have the following: Δ_M_ = 0.59 (*p* < 0.05) for community well-being, Δ_M_ = 1.2 (*p* < 0.001) for occupational well-being, Δ_M_ = 0.55 (*p* < 0.05) for physical well-being, Δ_M_ = 0.63 (*p* < 0.01) for psychological well-being, and Δ_M_ = 0.52 (*p* < 0.01) for overall well-being.

The only dimension for which a slightly different trend was found is economic well-being. In this case, the 2021 mean does not differ significantly from the 2020 mean, which in turn does not differ significantly from the 2022 mean. However, a significant positive difference emerges (*p* < 0.01) between the 2022 mean and that of 2021 (Δ_M_ = 0.56).

## 4. Discussion

The results of the present study indicate a general trend in the well-being levels of university students from the beginning of the pandemic to 2022. Compared to 2020, in 2021, there was a sharp decline in well-being, whereas in 2022 there was a trend towards an increase in the well-being levels.

This is particularly true for community well-being, physical well-being, psychological well-being, overall well-being and, above all, occupational well-being.

The results are in line with several studies conducted in recent years [37,38,39], which report a fluctuating trend in the well-being of university students.

The fluctuations in the perception of one’s well-being can be reinterpreted by recalling the results that emerged from a large national qualitative study [30]. This study on young university students involved writing reports on and describing the thoughts, emotions, and actions that characterized the period of the first lockdown, followed by the same after a year and, thereafter, the same after two years, during the same period of time that coincided with the first lockdown.

While many experienced anxieties during the first phase of the COVID lockdown, some saw it as an opportunity to enjoy their family relationships, rediscovering the pleasure of small things and being emotionally connected to their community [30,32]. These opportunities certainly protected the well-being of university students and prevented a worsening of their well-being. Di Napoli et al. [30] highlight how a feeling of solidarity, a sense of Italianness, emotional connection, trust, and a sense of responsibility towards others characterized the first phase of lockdown.

As widely underlined, the perception of a good state of well-being in the first lockdown seems to have been due to the feeling of belonging to the wider community [40]. Furthermore, Procentese et al. [41] highlighted how, during the first phase of the lockdown the perception of time changed in young people who declared that they experienced the lockdown period as a useful time to rethink their future plans.

In contrast, the year 2021 appears to be the one in which the young participants of the research felt a decline in well-being, which could be attributed to the distrust, tiredness, and impotence that they had probably developed due to the state of emergency.

Indeed, several studies supported the fact that after one year and on the occasion of the second lockdown, an increase in malaise was detected compared to the first lockdown, which could be understood by the hypothesis that the malaise was due to the long-term impact of the pandemic on the perception of well-being [42]. Furthermore, these studies found that after a year, the perception of social support was that it was no longer a predictor of well-being, compared to what had emerged during the first lockdown [43,44].

Regarding the increased perception of well-being in 2022, we can hypothesize that the effects of the resumption of daily activities (attendance at university, resumption of work, and reduction of restrictive measures for meeting places), in conjunction with the increase in the administration of vaccinations, had an impact on the well-being of young people.

Moreover, one interpretation of the observed increase in perceived well-being is connected to the ability to adapt, psychological flexibility [5], the use of resilience as a buffer against stress [45], and the feeling of trust in others that was kept alive by the opportunity to continue to interact through social media [46]. All of these abilities are essential skills for maintaining and protecting one’s psychological health.

Finally, one could also hypothesize, as Rosa [44] describes, that the increase in well-being could be connected to de-temporalization, or rather, to people’s ability to face critical and frustrating situations, thinking moment by moment. These data recorded in 2022 certainly encourage us to think about a recovery in the state of well-being, similar to that before the pandemic. However, these data still represent a partial result that deserves further in-depth research in the years to come.

## 5. Limitations and Future Proposals

The results of this study should be interpreted with the consideration of certain limitations. First, the research utilized convenience samples recruited through a non-probabilistic method, which limits the ability to generalize the findings.

Second, as regards the comparison of the well-being levels of 2020 and 2021 with those of 2022, the fact that the third sample is made up of 128 participants, which is a reduced number compared to the two other samples, may influence the validity of the analysis of variance and the subsequent post-hoc tests carried out. Therefore, further research with more numerically homogeneous samples is necessary to confirm the evidence found in the present study.

It is also important to point out that, although carried out over several years, this study does not have a longitudinal nature, as the participants recruited in the three groups are not the same. This may have led to a lower value of the results highlighted and implies the need for further studies.

Furthermore, future research is also necessary to have a greater temporal perspective of the trend in the well-being of university students after the advent of the pandemic. To gain a clearer understanding of the changes in well-being resulting from exceptional events like the COVID-19 pandemic, it will likely be necessary to conduct ongoing surveys over the coming years.

Despite these limitations, this study provides initial insights into subjective well-being during the peak of infections in 2020 and immediately after the return to normalcy amid preventive restrictions on personal freedom in 2021 and in the final moments of the state of emergency declared by the Italian government in 2022.

Based on our results, future studies can further assess the long-term effects of the COVID-19 pandemic and the lockdown experience on the well-being of university students and propose effective strategies for preventing and promoting people’s quality of life in general.

## 6. Conclusions and Practical Implications

The findings of this study reveal significant insights into the well-being of university students in Italy throughout the COVID-19 pandemic, highlighting notable trends from 2020 to 2022. These results underscore the importance of addressing the psychological challenges faced by university students during crises. Educational institutions should prioritize mental health support services, particularly during and after significant disruptions. Proactive measures could include enhancing access to counselling services, promoting peer support networks, and facilitating open discussions about mental health.

Moreover, as the data emphasize the influence of social connections on well-being, universities can foster community-building initiatives that encourage interaction among students, both online and in person. Activities that enhance interpersonal relationships and a sense of belonging may serve as protective factors against the adverse effects of future crises.

In conclusion, while the pandemic presented significant challenges, it also offered opportunities to reinforce the importance of mental health awareness and support systems within academic environments. By learning from these experiences, institutions can better prepare for future uncertainties, ensuring that student well-being remains a central focus.

## Figures and Tables

**Table 1 behavsci-14-00938-t001:** Demographic characteristics of the 2020, 2021, and 2022 samples.

	2020 n = 293		2021 n = 344		2022 n = 128	
Age	M = 20.9 (SD = 3.23)		M = 20.8 (SD = 2.52)		M = 20.8 (SD = 3.49)	
	**N**	**%**	**N**	**%**	**N**	**%**
Gender						
Female	234	80%	268	78%	101	79%
Male	59	20%	76	22%	27	21%
Sexual orientation						
Heterosexual	263	90%	310	90%	112	88%
Homosexual	12	4%	16	5%	8	6%
Other orientation	16	6%	18	5%	8	6%
Housing status						
With parents	265	90%	298	87%	116	91%
Alone	11	4%	10	3%	0	0%
With partners	2	1%	7	2%	4	3%
Another condition	13	5%	25	8%	8	6%

**Table 2 behavsci-14-00938-t002:** Standardized factor loadings (λ), inter-item reliability (R^2^), and Composite Reliability (CR) indices in the 2020, 2021, and 2022 samples.

Latent Variable	Item	2020 n = 293	2021 n = 344	2022 n = 128	
		λ (R^2^)	λ (R^2^)	λ (R^2^)	CR
Interpersonal well-being	Present	0.93 (0.90)	0.91 (0.85)	0.89 (0.82)	2020 → 0.89 2021 → 0.78 2022 → 0.82
	Future	0.86 (0.72)	0.68 (0.44)	0.78 (0.57)	
Community well-being	Present	0.93 (0.88)	0.90 (0.81)	0.87 (0.77)	2020 → 0.91 2021 → 0.83 2022 → 0.83
	Future	0.90 (0.80)	0.78 (0.62)	0.82 (0.68)	
Occupational well-being	Present	0.93 (0.89)	0.91 (0.89)	0.85 (0.76)	2020 → 0.91 2021 → 0.87 2022 → 0.77
	Future	0.90 (0.84)	0.85 (0.67)	0.72 (0.67)	
Physical well-being	Present	0.97 (0.96)	0.90 (0.81)	0.86 (0.74)	2020 → 0.92 2021 → 0.75 2022 → 0.80
	Future	0.88 (0.77)	0.64 (0.41)	0.77 (0.59)	
Psychological well-being	Present	0.93 (0.87)	0.90 (0.82)	0.84 (0.70)	2020 → 0.90 2021 → 0.79 2022 → 0.82
	Future	0.88 (0.77)	0.70 (0.49)	0.83 (0.69)	
Economic well-being	Present	0.94 (0.90)	0.92 (0.85)	0.87 (0.77)	2020 → 0.90 2021 → 0.82 2022 → 0.80
	Future	0.86 (0.75)	0.74 (0.51)	0.76 (0.59)	
Overall well-being	Present	0.96 (0.92)	0.90 (0.81)	0.81 (0.65)	2020 → 0.88 2021 → 0.87 2022 → 0.74
	Future	0.81 (0.66)	0.85 (0.63)	0.72 (0.51)	

**Table 3 behavsci-14-00938-t003:** ANOVA results for each dimension of well-being.

		DEV	df	VAR	F	*p*
Predictor: Year (2020; 2021; 2022)						
Interpersonal well-being	Between	52.01	2	26.00	8.79	0.000
	Within	2253.89	762	2.96		
	Total	2305.90	764			
Community well-being	Between	127.65	2	63.82	16.15	0.000
	Within	3012.15	762	3.95		
	Total	3139.80	764			
Occupational well-being	Between	183.71	2	91.86	37.00	0.000
	Within	1891.54	762	2.48		
	Total	2075.25	764			
Physical well-being	Between	95.49	2	47.74	14.31	0.000
	Within	2541.48	762	3.34		
	Total	2636.97	764			
Psychological well-being	Between	119.18	2	59.59	16.34	0.000
	Within	2779.72	762	3.65		
	Total	2898.90	764			
Economic well-being	Between	29.55	2	14.78	4.65	0.010
	Within	2423.51	762	3.18		
	Total	2453.07	764			
Overall well-being	Between	98.06	2	49.03	17.03	0.000
	Within	2194.20	762	2.88		
	Total	2292.26	764			

**Table 4 behavsci-14-00938-t004:** Post hoc test for comparing the mean levels of well-being over the three years.

Outcome	Year			2020			2021	
		M	SD	Δ_M_	*p* Tukey	*p* Dunnet	Δ_M_	*p* Tukey
Interpersonal well-being	2020	7.59	1.43	/	/	/	/	/
	2021	7.02	1.81	−0.567	0.000	0.000	/	/
	2022	7.39	2.04	−0.202	0.444	0.509	0.365	0.101
Community well-being	2020	5.92	1.79	/	/	/	/	/
	2021	5.04	2.07	−0.885	0.000	0.000	/	/
	2022	5.63	2.18	−0.294	0.343	0.284	0.591	0.012
Occupational well-being	2020	7.15	1.37	/	/	/	/	/
	2021	6.30	1.65	−0.852	0.000	0.000	/	/
	2022	7.49	1.80	0.344	0.099	0.075	1.20	0.000
Physical well-being	2020	7.38	1.49	/	/	/	/	/
	2021	6.62	1.96	−0.759	0.000	0.000	/	/
	2022	7.18	2.14	−0.206	0.535	0.470	0.552	0.010
Psychological well-being	2020	6.45	1.72	/	/	/	/	/
	2021	5.60	1.95	−0.846	0.000	0.000	/	/
	2022	6.23	2.21	−0.220	0.522	0.458	0.626	0.005
Economic well-being	2020	6.29	1.60	/	/	/	/	/
	2021	6.10	1.86	−0.193	0.361	0.301	/	/
	2022	6.66	1.96	0.367	0.128	0.098	0.560	0.007
Overall well-being	2020	7.13	1.46	/	/	/	/	/
	2021	6.36	1.76	−0.776	0.000	0.000	/	/
	2022	6.88	1.97	−0.254	0.334	0.276	0.521	0.009

## Data Availability

Data are contained within the article.

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
