# Peer review of "COVID-19 and University Students’ Well-Being: An Ecological and Multidimensional Perspective on Post-Pandemic Effects"

_behavsci, 2024, doi:10.3390/bs14100938_

Round 1

Reviewer 1 Report

Comments and Suggestions for Authors

Nice work. Please see my comments in the attached pdf files

Comments on the Quality of English Language

Minor editing of the language usedi in the text is required.

Author Response

REVIEW 1

TITLE AND ABSTRACT The title needs to be abbreviated so that it can neatly summarize the objective of the paper. Perhaps something along the lines of An Ecological and Multidimensional Perspective on Pre and Post-Pandemic Effects

A: As suggested, we shortened the title.

The abstract is coherent and succinctly encapsulates the main objective of the article offering information on the research methodology, instruments and key results of the survey reported in the paper. However, there are some minor spelling (line 1) errors that are still pending editing.

A: We corrected the typos in the abstract

INTRODUCTION The introduction the topic of wellbeing and its subsequent decline due to the COVID-19 advent is treated as a common issue for the general population. It is not obvious why it is important to examine it in the context of higher education and how students have experienced it both pre- and post-pandemically. In fact, taking into account that we are already in 2024, this paper appears a bit untimely as a number of studies have already addressed the issue from multiple perspectives adopting a number of different research methodologies. Line 53 mentions something about a salutogenetic perspective without making elaborating on it more Line 72 I would have like the I COPPE model which is I assume- the main theoretical framework for this paper to be further analyzed and its adoption be justified.

A: We modified the introduction and made it more explicit why we have placed the focus on the well-being of university students. We also added more details explaining why we have chosen the I COPPE model as the main theoretical framework.

LITERATURE REVIEW There is no section of literature review on the topic there are studies devoted to the topic.

A: We added a section that addresses the well-being of university students, in the wake of the covid-19 pandemic.

MATERIALS AND RESULTS Lines 89-91: If the study was meant to be a longitudinal one tracing wellbeing levels of a group of university students, then the same number of students would be required in all three points where data were collected so that cross-comparisons could be valid and conclusions drawn can be reliable. I think this is one of the key methodological mistakes in this study as also mentioned in the limitations that also jeopardizes the reliability of the findings.

A: We further highlighted, in the limitations, that the study does not have a longitudinal nature and therefore the results need to be further explored in the future.

Lines 106-113; Is the I COPPE scale a measure for general levels of wellbeing or academic wellbeing? If the former, was it suitable for use within the educational setting of higher education? What made the researchers choose this scale and, not others? that have to with academic wellbeing as experienced by university students?

A: In the introduction, we explained more clearly the usefulness of the I COPPE model and its previous applications with university students.

Results Section: In comparing levels of wellbeing of different sample sizes across three periods in time seems to be unreliable! The trend observed is what could be expected and agrees with findings of previous studies but unfortunately no previous research is mentioned!

A: In the discussions, we extended the coverage of previous studies, in line with ours, to give greater strength to the results found.

DISCUSSION AND CONCLUSIONS Some of this research is mentioned in the Discussion section, so there needs to be a lit review section to see if there is a need for further research as researchers of this paper try to convince us.

A: In the introduction, we added a detailed analysis of studies that have addressed the well-being of university students in recent years. And based on what was reported, we specified what future research should focus on.

No practical applications are suggested e.g. interventions in higher education to boost students' wellbeing in highly critical emergency periods such as the pandemic.

A: We added a section reporting the practical implications of the study.

I consider the paper to be an untimely contribution to the topic of university students' wellbeing that should have been published in the COVID-19 years. Two major methodological mistakes that render the results of the study to be totally unreliable are: (1) the comparison of different sample sizes across the three time points and (2) the selection of I COPPE scale as a measure of university students' wellbeing. No practical implications are discussed on the promotion and support of students' wellbeing in higher education during critical times.

A: In the various previous points, we have tried to resolve these critical issues highlighted by the reviewer.

Reviewer 2 Report

Comments and Suggestions for Authors

The English language used in this paper needs to be improved. In addition, here are some other points for consideration:

Page 1, line 11: “February 202” should be “February 2022”.

Page 1, line 13: Consider replacing “One of the categories …” with “One group of people …”

Page 1, line 17: “765 participant …” don’t start a sentence with a number. Instead say “A sample of 765 participants (M = 21 years, SD = 2.87) …”

Page 1, line 30: There is a mistake with “as is widely l known”

Page 1: For the Introduction section, the first three paragraphs should be just one paragraph.

Page 2, line 48: The sentence “In particular, this perception of insecurity …” does not seem to follow from the previous sentence. ‘Insecurity’ is not the same as ‘uncertainty and loneliness.’
Page 2, line 53: ‘protec’ should be ‘protect.’

Page 2, line 64: “Starting from these initial …” should be “Starting from the initial …”

Page 2, the third paragraph in the Method section is not needed as the data are contained in Table 1.

Page 5, line 154: What does “wave-like trend” mean?

The term “highly significant” is not needed. It should just be “statistically significant.”

Page 7, line 180: Should the phrase “… towards an increase in scores” be “… towards an increase in wellbeing”?

Page 8, lines 216-220: The sentence “Moreover, it could be interpreted that the increase in the perception of well-being, …” was poorly written. It needs to be rewritten. As a suggestion, it could commence with: “Moreover, one interpretation of the observed increase in perceived well-being is ...”

Comments on the Quality of English Language

The English language used in this paper needs to be improved.

For example, the sentence “Regarding reliability, as reported in Table 2., the Composite Reliability (CR) indices demonstrated a good level (>.7) for all dimensions of the I COPPE scale, in all three samples” could be rewritten as “The Composite Reliability (CR) indices demonstrated a good level of reliability (>.7) for all dimensions of the I COPPE scale in all three samples (see Table 2)”.

As another example, the sentence "As described by Migliorini et al., (2021) and Di Napoli et al., (2021), the first phase of the lockdown, albeit as a new condition, was experienced as being full of anxiety about what was happening, but also as an opportunity to enjoy one's family relationships, rediscovering the pleasure of small things, and finding oneself emotionally connected to one's community" could be rewritten as "While many experienced anxiety during the first phase of the COVID lockdown, some saw it as an opportunity to enjoy one's family relationships, rediscovering the pleasure of small things, and being emotionally connected to one's community.

".

Author Response

REVIEW 2

Comments and Suggestions for Authors The English language used in this paper needs to be improved.

A: We revised the English throughout the manuscript.

In addition, here are some other points for consideration: Page 1, line 11: “February 202” should be “February 2022”.

A: We corrected it.

Page 1, line 13: Consider replacing “One of the categories …” with “One group of people …”

A: We corrected it.

Page 1, line 17: “765 participants …” don't start a sentence with a number. Instead say “A sample of 765 participants (M = 21 years, SD = 2.87) …”

A: We corrected it.

Page 1, line 30: There is a mistake with “as is widely known”

A: We corrected it.

Page 1: For the Introduction section, the first three paragraphs should be just one paragraph.

A: We merged the three paragraphs into one.

Page 2, line 48: The sentence “In particular, this perception of insecurity …” does not seem to follow from the previous sentence. 'Insecurity' is not the same as 'uncertainty and loneliness.'

A: We corrected it.

Page 2, line 53: 'protec' should be 'protect.'

A: We corrected it.

Page 2, line 64: “Starting from these initial …” should be “Starting from the initial …”

A: We corrected it.

Page 2, the third paragraph in the Method section is not needed as the data are contained in Table 1.

A: We corrected the paragraph, reporting as little data as possible and referring the reader to Table 1.

Page 5, line 154: What does “wave-like trend” mean?

A: We meant to say “a wave-like trend.” We corrected that.

The term “highly significant” is not needed. It should just be “statistically significant.”

A: We corrected it.

Page 7, line 180: Should the phrase “… towards an increase in scores” be “… towards an increase in wellbeing”?

A: Yes. We corrected it.

Page 8, lines 216-220: The sentence “Moreover, it could be interpreted that the increase in the perception of well-being, …” was poorly written. It needs to be rewritten. As a suggestion, it could begin with: “Moreover, one interpretation of the observed increase in perceived well-being is …”

A: We corrected it.

Comments on the Quality of English Language The English language used in this paper needs to be improved.

A: We revised the English throughout the manuscript.

For example, the sentence “Regarding reliability, as reported in Table 2., the Composite Reliability (CR) indices demonstrated a good level (>.7) for all dimensions of the I COPPE scale, in all three samples” could be rewritten as “The Composite Reliability (CR) indices demonstrated a good level of reliability (>.7) for all dimensions of the I COPPE scale in all three samples (see Table 2)”.

A: We accepted the reviewer's suggestion and corrected the sentence.

As another example, the sentence "As described by Migliorini et al., (2021) and Di Napoli et al., (2021), the first phase of the lockdown, albeit as a new condition, was experienced as being full of anxiety about what was happening, but also as an opportunity to enjoy one's family relationships, rediscovering the pleasure of small things, and finding oneself emotionally connected to one's community" could be rewritten as "While many experienced anxiety during the first phase of the COVID lockdown, some saw it as an opportunity to enjoy one's family relationships, rediscovering the pleasure of small things, and being emotionally connected to one's community

A: We accepted the reviewer's suggestion and corrected the sentence.

Reviewer 3 Report

Comments and Suggestions for Authors

The article is extremely interesting, it reveals the impact of the COVID-19 pandemic on the well-being of university students and contributes to future research into the mid- and long-term effects of the pandemic.

Still, I have a few minor suggestions.

1- The title is too long. I suggest it should be more summarise and objective.

2- There are a few typos, por example, "In February 202," (line 11),  I suggest that the authors review the entire work and look for possible gaps;

3- I suggest that the discussion of the concept of well-being should be included in the article, not only because it is a central issue of the investigation but also because the complexity of its definition and the multiplicity of approaches cannot be summed up by referring to the use of the I COPPE scale and its multidimensional vision of well-being;

4- The article presents the results of a research carried out between 2020 and 2022 and not "from the beginning of the pandemic to today" as described (line 177). I recommend a correction.

Author Response

REVIEW 3

1- The title is too long. I suggest it should be more summary and objective.

A: We shortened the title.

2- There are a few typos, for example, "In February 202," (line 11), I suggest that the authors review the entire work and look for possible gaps;

A: We fixed the error and revised the entire manuscript.

3- I suggest that the discussion of the concept of well-being should be included in the article, not only because it is a central issue of the investigation but also because the complexity of its definition and the multiplicity of approaches cannot be summed up by referring to the use of the I COPPE scale and its multidimensional vision of well-being;

A: We expanded both the introduction and discussions, looking more deeply into the topic of university students’ well-being after the pandemic.

4- The article presents the results of a research carried out between 2020 and 2022 and not "from the beginning of the pandemic to today" as described (line 177). I recommend a correction.

A: We corrected it.

Round 2

Reviewer 1 Report

Comments and Suggestions for Authors

Well done for your work!

Reviewer 2 Report

Comments and Suggestions for Authors

Perhaps for the first sentence in the Conclusion section, replace the word 'significant' with 'important'?